# Density Resolved Wave Packet Spreading in Disordered Gross-Pitaevskii Lattices

Yagmur Kati[1,2,5*], Xiaoquan Yu[3,4], Sergej Flach[1,2,5]

**1** Center for Theoretical Physics of Complex Systems, Institute for Basic Science (IBS), Daejeon 34126, Korea
**2** Basic Science Program, Korea University of Science and Technology (UST), Daejeon 34113, Korea
**3** Graduate School of China Academy of Engineering Physics, Beijing 100193, China
**4** Department of Physics, Centre for Quantum Science, and Dodd-Walls Centre for Photonic and Quantum Technologies, University of Otago, Dunedin 9010, New Zealand.
**5** New Zealand Institute for Advanced Study, Centre for Theoretical Chemistry and Physics, Massey University, Auckland 0745, New Zealand.
* ygmrkati@gmail.com

November 17, 2021

## Abstract

We perform novel energy and norm density resolved wave packet spreading studies in the disordered Gross-Pitaevskii (GP) lattice to confine energy density fluctuations. We map the locations of GP regimes of weak and strong chaos subdiffusive spreading in the 2D density control parameter space and observe strong chaos spreading over several decades. We obtain a renormalization of the ground state due to disorder, which allows for a new disorder-induced phase of disconnected insulating puddles of matter due to Lifshits tails. Inside this Lifshits phase, the wave packet spreading is substantially slowed down.

# 1 Introduction

Disorder is inevitable naturally in all materials due to the impurities or defects caused by external fields. Interacting wave dynamics in disordered and incommensurate lattice structures are actively studied due to its complex properties [1]. Interacting waves can be modeled with nonlinear terms approximating true quantum many-body physics of interacting bosons. In the absence of interactions, Anderson localization (AL) leads to exponential localization of eigenstates and a coherent halt of wave packet spreading [2]. The complete suppression of wave propagation has been manifested by a bevy of experimental observation; including localization of light waves [3], photonic crystals [4], sound waves [5], microwaves [6], and atomic matter waves [7, 8]. In the presence of nonlinear wave interaction terms, delocalization can arise and ultimately lead to chaotic dynamics, which destroys Anderson localization through incoherent spreading [9–11]. Notably, this phenomenon was also studied experimentally with ultracold atomic gases [12]. As interesting is the fact that experimental efforts are limited by the time of atomic condensate control, which - assuming a natural dimensionless time scale of order $t = 1$ - yields the largest realizable times of the order of $t \sim 10^4$ (the typical natural time scale is of the order of 10ms, and condensate control usually extends up to 10s). At these experimental time horizons, the observation of the onset of incoherent spreading was possible, but a quantitative assessment of this process was not. Theoretical simulations can achieve $t \sim 10^8 - 10^9$ [1], and in special settings of unitary maps with discrete-time quantum walks are reaching unprecedented times $t \sim 10^{12}$ [13]. These numbers demonstrate the rare opportunity for computational studies being the superior testbed of the first choice.

Let us recap nonlinear wave packet spreading in simple terms. At the initial time $t = 0$, a wave packet is assumed to have a compact distribution of finite norm $\mathcal{A}$ and energy $\mathcal{H}$, which extends over $L$ lattice sites. Deprived of its nonlinear terms, the system manifests AL through exponentially localized eigenstates, of which roughly $L$ are excited by the wave packet. Transforming into normal mode space yields a harmonic oscillator equation for each AL eigenstate, with the nonlinear terms inducing a short-range coupling between them. Assume that this dynamical system will be nonintegrable, characterized by nonzero Lyapunov coefficients, and evolving chaotically in time. The consequent phase decoherence of the normal modes removes the basis of existence for AL, and normal modes in the boundary layer at the edges of the wave packet will be incoherently excited. The wave packet will spread, and $L^2(t) \sim t^\alpha$ increase in time. The assumption of complete dephasing of all AL normal modes yields the strong chaos regime $\alpha_s = 1/2$ for nonlinearity originating from two-body interactions [11]. However, the computationally tested asymptotic weak chaos regime was observed to yield $\alpha_w = 1/3$. This result can be derived assuming that the probability of a normal mode being resonant and chaotic is proportional to energy and/or norm density in the wave packet [11] (notably this assumption results in dependence of both $\alpha_w$ and $\alpha_s$ on the lattice dimension and different choices of $N$-body interactions [1]). Whether the observed weak chaos spreading is asymptotic or will slow down, has been a topic of debates and discussions [14], with still no answer in sight. What was confirmed in computations, is the potentially long-lasting inter-

mediate strong chaos regime [15] - but notably only for systems with one integral of motion (energy) [15].

The spreading wave packet is assumed to thermalize on time scales shorter than the one on which it spreads. For systems conserving energy only, the energy density $h(t) = \mathcal{H}/L(t)$ corresponds to some inverse temperature $\beta(t)$, and while spreading $h(t \to \infty) \to 0$ and thus $\beta(t \to \infty) \to \infty$, so the packet spreads and cools down. A thermal Gross-Pitaevskii (GP) wave packet however conserves, in addition, the total norm $\mathcal{A}$ and must be characterized by two energy $h(t)$ and norm $a(t) = \mathcal{A}/L(t)$ densities which are related to some inverse temperature $\beta(t)$ and chemical potential $\mu(t)$ [16]. The spreading dynamics then correspond to moving along a line in the density parameter space $\{a, h\}$ which connects an initial point $\{a_0, h_0\}$ with the origin. At variance to systems with only one integral of motion, a GP lattice supports non-Gibbs phases [17], and hence the outcome will depend on the chosen line, including heating upon spreading and possibly reaching infinite temperatures at finite values of $L$. Computational studies of GP wave packet dynamics [15] involved disorder averaging which was controlling $a_0$ but not $h_0$, therefore averaging over fans of lines in the density parameter space.

In this work, we will unfold the density-resolved dynamics, which allows us to finally identify a clean strong chaos regime, and the potential slowing down Lifshits phase regimes. We observe strong chaos and map strong and weak chaos in the density parameter space, including a localization regime coined Lifshits phase (LP) due to a disorder-induced ground state renormalization.

## 2    Model definition

We consider the disordered Gross-Pitaevskii chain on $N$ sites with Hamiltonian

$$\mathcal{H} = \sum_{\ell=1}^{N} \left[ -J(\psi_\ell^* \psi_{\ell+1} + \psi_\ell \psi_{\ell+1}^*) + \epsilon_\ell n_\ell + \frac{g}{2} n_\ell^2 \right] , \tag{1}$$

where $\psi_\ell = \sqrt{n_\ell} e^{i\phi_\ell}$ are complex scalars, and the integer $\ell$ enumerates the lattice sites. $\epsilon_\ell$ is a quenched uncorrelated on-site random potential taken to be uniformly distributed within a box of size $W$: $\epsilon_\ell \in [-\frac{W}{2}, \frac{W}{2}]$. $W$ is a measure of the disorder strength, $J$ is the tunneling amplitude between neighboring sites, and $g > 0$ is the nonlinearity parameter resulting from e.g. the repulsive two-body interaction between atoms. Energy (and the inverse of time) can be measured in units of $J$ which leaves us with two parameters: $W, g$. Uniform rescaling of the norm $|\psi_\ell|^2$ is used to tune the nonlinearity parameter into $g = 1$.

For $g = 0$, the system (1) is reduced to an eigenvalue problem using $\psi_\ell(t) = \Psi_\ell e^{-i\lambda t}$: $\lambda \Psi_\ell = \epsilon_\ell \Psi_\ell - (\Psi_{\ell+1} + \Psi_{\ell-1})$. It follows $|\lambda_\nu| \le 2 + W/2$, with all eigenvectors $\Psi_{\ell,\nu}$ being exponentially localized in space for any $W \ne 0$ [2]. The localization length $\xi(\lambda_\nu)$ takes its largest value $\xi_0 \approx 96/W^2$ for $\lambda_\nu = 0$ [18]. The typical core size of a corresponding eigenvector at energy $\lambda = 0$ fluctuates with an average of the order of $V_{loc} \approx 3\xi_0$ [11]. This follows from the numerical observation that the participation number of such a normalized eigenstate $P_\nu = \sum_\ell |\Psi_\ell|^4 \approx 1.5\xi_\nu$, whose random amplitude fluctuations result in another factor of 2 for the size of the state (which accounts for a site amplitude being either large or small with equal probability). The localization volume hosts $V_{loc}$ eigenstates with amplitudes of order $V_{loc}^{-1/2}$.

That group of significantly overlapping eigenstates has $V_{loc}$ eigenvalues which are separated from each other by the average level spacing $d(W) = (4 + W)/V_{loc}$.

With $J \equiv g \equiv 1$ the final equations of motion $\dot\psi_\ell = \partial\mathcal{H}/\partial i\psi_\ell^*$ read

$$i\dot\psi_\ell = \epsilon_\ell\psi_\ell + n_\ell\psi_\ell - (\psi_{\ell+1} + \psi_{\ell-1}). \tag{2}$$

The above dynamics conserves the total norm $\mathcal{A} \equiv \sum_\ell n_\ell$ and the total energy $\mathcal{H}$. The partition function

$$\mathcal{Z} = \int_0^\infty \int_0^{2\pi} \prod_\ell d\phi_\ell dn_\ell \exp[-\beta(\mathcal{H} + \mu\mathcal{A})], \tag{3}$$

where $\beta$ and $\mu$ are Lagrange multipliers associated with the total energy and the total norm, respectively.

Let us recap the equilibrium properties of the GP model assuming nonzero overall densities $h = \mathcal{H}/N$ and $a = \mathcal{A}/N$. For the ordered case $W = 0$ the microcanonical ground state

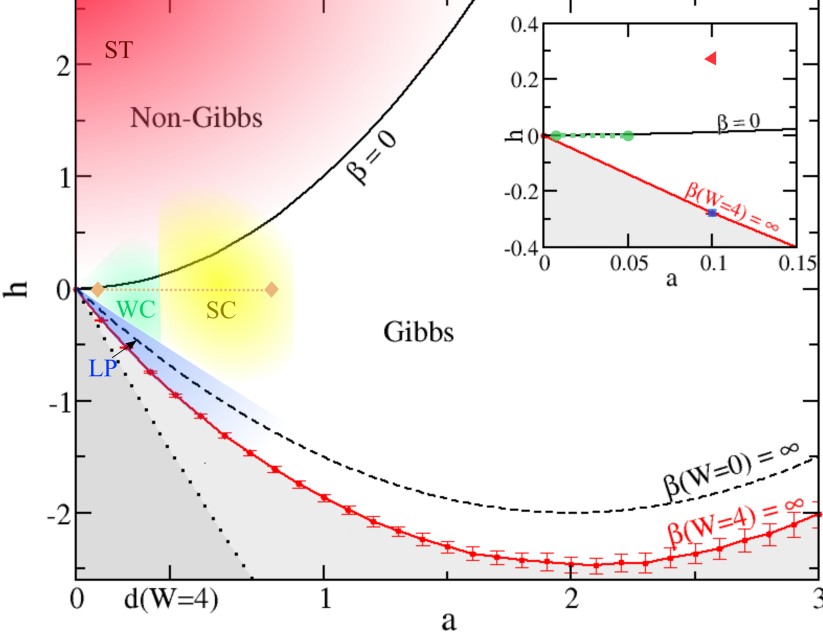

Figure 1: Phase diagram of the microcanonical GP system. Black dashed line - ground state $h = -2a + a^2/2$ for $W = 0$. Red filled circles - the renormalized ground state for $W = 4$. The red solid line connects the data and guides the eye. The black solid line $h = a^2$ corresponds to infinite temperature $\beta = 0$ for any strength of disorder. The four shaded areas correspond to strong chaos (SC), weak chaos (WC), self-trapping (ST), and Lifshits phase (LP). The tick label $d(W = 4)$ marks the position where the norm density equals the average level spacing $d$. The dotted black line represents the absolute minimum energy line $h = -(2 + W/2)a + a^2/2$ reachable for finite systems. SC spreading ($W = 4$, Fig.4) is shown at $t = 0$ and $t = 10^6$ with orange diamonds connected by the orange dotted line path. Inset: WC spreading ($W = 4$, Fig.3) is shown at $t = 0$ and $t = 10^9$ with green circles, connected with the green dotted line path. We used $a(t) = a(0)/L(t)$ by assuming the wave packet is uniform while spreading at finite $T$. The initial states for ST and LP ($W = 4$, Fig.2,5) are shown as a red triangle, and blue square, respectively.

line $h = -2a + a^2/2$ equals the grand-canonical zero temperature $\beta = \infty$ line, and the line $h = a^2$ corresponds to the infinite temperature line $\beta = 0$ [17] (see Fig.1). Pairs of densities in the Gibbs range $-2a + a^2/2 \leq h \leq a^2$ are addressable by pairs of positive inverse temperature and chemical potential $\{\beta, \mu\}$. The entire non-Gibbs density range $h > a^2$ is not captured by a positive temperature, while negative temperature assumptions lead to a divergence of partition functions, and microcanonical dynamics show strong deviations from expected ergodic behavior including self-trapping [17,19]. The Gibbs-nonGibbs separation line in the density space was recently shown to persist for entire classes of generalized GP lattice equations as well as their Bose-Hubbard quantum counterparts, for any lattice dimension, and in the presence of disorder [20]. Remarkably the addition of disorder for (1) leaves the infinite temperature line $h = a^2$ invariant [16].

## 3 Ground state renormalization

The zero-temperature ground state line of the ordered case $h = -2a + a^2/2$ is renormalized in the presence of disorder. This happens in the regime of small norm density (i.e. weak nonlinearity) $a < 1$ due to the presence of Lifshits states which are sparsely distributed AL eigenstates with eigenvalues close to the bottom of the AL spectrum, i.e. their distance from the bottom $\Delta_\lambda = \lambda + 2 + W/2 \ll 1$. Such Lifshits states exist due to rare disorder fluctuations with $\epsilon_l + W/2 < \Delta_\lambda$ over a simply connected chain segment of length $L = \pi/\sqrt{2\Delta_\lambda}$. The average distance between such regions $d_L \approx (W/\Delta_\lambda)^L$. As a result, one can expect a set of disjoint puddles of norm distribution in real space for small norm density $a$. Note also that for any finite system the ground state is bounded by $h = -(2 + W/2)a + a^2/2$ which is generated by the disorder realization $\epsilon_\ell = -W/2$.

Contrary, in the large norm density limit (i.e. for strong nonlinearity) the ground state correction becomes weak since the nonlinear terms $a^2/2$ are of leading order and disorder has a minor impact.

In order to numerically compute the ground state, we note that $\psi_\ell$ can be gauged into real variables as all the phases $\phi_\ell = \phi_{\ell'}$ to minimize the Hamiltonian (1). The remaining task is to minimize a real function $\mathcal{H}$ for real variables $\psi_\ell$ for a given disorder realization. We choose an initial set of $\psi_\ell$ under the constraint $Na = \sum_\ell \psi_\ell^2$. We define a window of $l_w = 5$ adjacent sites and minimize the energy varying the amplitudes on these adjacent sites using the Nelder-Mead simplex algorithm [21]. As the algorithm changes the total norm in general, we perform a homogeneous renormalization of all amplitudes $\psi_\ell$ to restore the required norm density $a$. We then shift the window by one lattice site and repeat the procedure, until the whole lattice with $N$ sites has been covered by minimization windows. The procedure is repeated 10 times, after which full convergence is obtained. The chemical potential

$$\mu = \epsilon_\ell + gn_\ell - (\psi_{\ell+1} + \psi_{\ell-1})/\psi_\ell \tag{4}$$

is defined through local relations and yields a ratio of the standard deviation to mean which is less than $10^{-3}$, indicating the quality of our ground state computation. Finally, we repeat the procedure for 100 different disorder realizations and compute the average ground state energy density $h$ and its standard deviation. The result is shown as red solid circles in Fig.1 with their standard deviation for $N = 1000$, and $W = 4$. Optimizing small parts of the lattice at a time immensely reduced the computational time, enabling us to reach the ground states for larger system sizes, e.g. $N = 50000$.

# 4 Initial conditions

We prepare a wave packet on $L \approx V_{loc}$ consecutive sites in the center of a disordered lattice (see table 1).

Table 1: Length of initial wave packets $L$ for chosen $W$

| W | 1 | 2 | 3 | 4 | 6 | 8 |
|---|---|---|---|---|---|---|
| L | 361 | 91 | 37 | 21 | 10 | 6 |

For $a^2/2 - 2a \leq h \leq a^2/2 + 2a$, we choose an initial state with a homogeneous norm distribution $\psi_\ell = \sqrt{a}e^{i\phi_\ell}$. We fix the phase differences

$$\Delta\phi = \phi_\ell - \phi_{\ell+1} = \arccos\left(\frac{h}{2a} - \frac{a}{4}\right). \tag{5}$$

We then adjust the phase on one of the sites to tune the total energy such that $\mathcal{H} = Lh$ (and disregard disorder realizations for which the adjustment can not be realized).

For $h > a^2/2 + 2a$ or $h < a^2/2 - 2a$, we use the ground state renormalization method, replacing the original energy with $|\mathcal{H} - Lh|$ and strictly varying only the amplitudes of the wave function on the $L$ sites of the wave packet. With that, we can prepare density resolved wave packets which are characterized by a pair of initial density values $\{a_0, h_0\}$, and a corresponding point in the phase diagram Fig.1. A spreading wave packet is characterized by a pair of time-dependent densities $a(t)$ and $h(t)$ moving along a straight line connecting the initial phase diagram point with the origin.

Parametrizing the line as $h(t) = ca(t)$ and assuming $a(t) \ll 1/g$, we approximate the partition function by its interaction-free limit $g = 0$ and derive $h = -\frac{\mu}{\beta}(\mu^2 - 4)^{-1/2} + \frac{1}{\beta}$ and $a = \frac{1}{\beta}(\mu^2 - 4)^{-1/2}$ for the ordered case. The line parametrization then finally yields $\beta = -\frac{2c}{a(4-c^2)}$ and $\mu = -\frac{4+c^2}{2c}$.

We can expect a number of qualitatively different spreading outcomes for the disordered case depending on the choice of the initial density values. If $h_0 > 0$ and the initial point is in the Gibbs phase, the origin-connecting line will cross the infinite temperature line. Therefore the wave packet will heat up to infinite temperatures, enter the non-Gibbs phase, and is expected to show features of self-trapping, fragmentation, and condensation. For $h_0 = 0$ and $c = 0$ it follows $h(t) = 0$, the temperature will gradually increase and will reach infinite values at infinite times. If $-2a_0 + a_0^2/2 < h_0 < 0$ (i.e. $c < 0$), the wave packet may heat up to some finite temperature, but will asymptotically gradually cool down and reach potentially zero temperature upon approaching the origin at infinite times. Finally, if $h_0 < -2a_0 + a_0^2/2$, the wave packet can evolve in the Lifshits phase.

# 5 Computational details

We integrate Eq.2 using a symplectic method SBAB$_2$ [22, 23] for several different disorder realizations with time step size $\Delta t = 0.1$ unless mentioned otherwise.

The wave spreading is characterized by two main ingredients. The second moment $m_2 \equiv \sum_\ell (l - \bar{l})^2 |\psi_\ell(t)|^2/\mathcal{A}^2$ measures the width of the wave packet. Here $\bar{l} = \sum_\ell l|\psi_\ell(t)|^2/\mathcal{A}^2$ is the

center of the wave packet. Participation number $P \equiv \mathcal{A}^2/\sum_\ell |\psi_\ell|^4$ characterizes the bulk of the wave-packet and allows us to compute the compactness index $C \equiv P^2/m_2$. If $C \sim 1$, then the wave packet is close to a thermal distribution, while $C \ll 1$ indicates either fragmentation of the wave packet into a set of essentially disjoint pieces or a part of the wave packet staying localized with another part spreading [1]. In order to observe whether asymptotic subdiffusion $m_2 \sim t^\alpha$ is recorded, we compute

$$\alpha(t) \equiv \frac{d\langle \log_{10} m_2 \rangle}{d \log_{10} t}. \tag{6}$$

The obtained function $\alpha(t)$ is additionally smoothened using a Hodrick-Prescott filter with a standard deviation of the output of less than 2% [24].

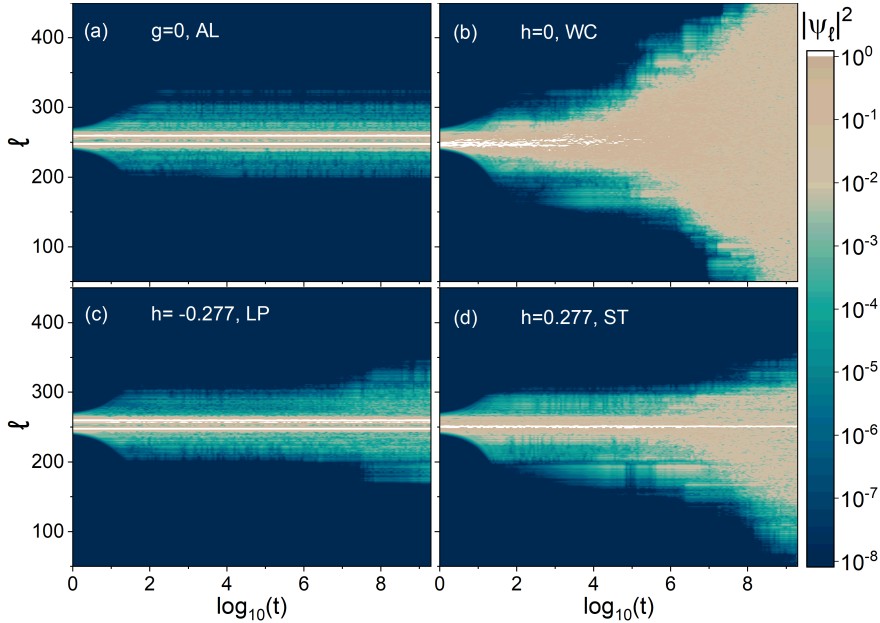

Figure 2: Different regimes of density resolved wave packet spreading. The evolution of the norm density $|\psi_\ell|^2$ is plotted versus $\log_{10} t$. (a) Anderson localization (AL): $g = 0, h = -0.277$. (b) weak chaos (WC): $g = 1, h = 0$. (c) Lifshits phase (LP): $g = 1, h = -0.277$. (d) self-trapping (ST): $g = 1, h = 0.277$ (here $\Delta t = 0.05$). For all cases $a = 0.1, W = 4$, and one and the same disorder realization are used here for all regimes.

# 6  Observation of weak and strong chaos

In the absence of nonlinear interactions $g = 0$, a wave packet will evolve in time without appreciable spreading. We plot the evolution of its norm density versus space and time in Fig.2(a). After some short initial dynamics during which the field established exponentially localized tails in space, the wave packet evolution essentially halts, signaling Anderson localization.

To avoid non-Gibbs dynamics or a subsequent cooling of the wave packet, we choose $h_0 = 0$. A localized Anderson mode of the linear system will be interacting with $d(W)$ neighboring other modes in the presence of nonlinear interactions. For $a > d$, that interaction is expected

to exhibit strong resonances and lead to efficient norm mixing among the participating modes [1]. Upon further wave packet spreading, the norm $a(t)$ will decrease, and cross over into the asymptotic regime of weak chaos $a < d$. The evolution of the norm density of a wave packet in the weak chaos regime is plotted versus space and time in Fig.2(b). At variance to the case of Anderson localization Fig.2(a), the wave packet grows in size with increasing time.

The weak chaos regime is characterized by $m_2 \sim t^{1/3}$, as reported in Ref. [11]. We confirm these findings in our computations as shown in Fig.3. We launch wave packets for various values of $W$ and $a(t = 0) < d(W)$. Both second moment $m_2(t)$ and participation number $P(t)$ indicate subdiffusive spreading (Fig.3 (a,b)). The compactness index $C(t \sim 10^8) \approx 3$, as expected for a thermalized system (Fig.3(c)). A quantitative evaluation of the time dependence of the exponent $\alpha(t)$ indicates its asymptotic convergence to $\alpha(t \to \infty) \approx 1/3$.

In order to observe intermediate strong chaos $m_2 \sim t^{1/2}$, we simply need to increase the initial norm density $a > d$. The evolution of the norm density of a wave packet in the strong chaos regime is qualitatively similar to the weak chaos evolution Fig.2(b). The wave packet will spread in the strong chaos regime until some time at which $a(t) \approx d$, which will mark the crossover to weak chaos. That crossover was observed for Klein-Gordon lattices in Ref. [15]. However the attempt to observe strong chaos and the crossover to weak chaos for the GP lattice failed [15]. The main reason is that the initial norm density $a$ of the wave packet was controlled, but the initial energy density $h$ was fluctuating when choosing and averaging over different disorder realizations [25, 26]. As a result, qualitatively different regimes of strong chaos, self-trapping, and Lifshits phases were mixed into one curve. Interestingly the results in Ref. [11] were obtained for single-site excitations which *did* control both norm and energy, but the strong chaos regime was not observed because single-site excitations get self-trapped. Fig.4 shows our results when controlling both initial densities and performing density-resolved spreading studies. Both second moment $m_2(t)$ and participation number $P(t)$ indicate subdiffusive spreading (Fig.4 (a,b)). The compactness index $C(t \sim 10^8) \approx 3$, as expected for a thermalized system (Fig.4(c)). We observe strong chaos with $\alpha(t)$ reaching the value $1/2$ and keeping this value for several decades in time, before slowly decaying, presumably to its asymptotic value $1/3$.

# 7 Lifshits phase and self-trapping

In the Lifshits phase, the spreading of the wave packet is slowing down dramatically. The evolution of the norm density of a wave packet in the Lifshits phase is plotted versus space and time in Fig.2(c). It shows very little difference to the linear case of Anderson localization in Fig.2(a). The second moment grows slightly (blue line Fig.5(a)), while the participation number is essentially frozen (blue line Fig.5(b)). The compactness index drops with increasing time (blue line Fig.5(a)). The derivative $\alpha(t)$ (blue line Fig.5(d)) shows a slow increase.

In the non-Gibbs regime, the GP dynamics turn non-ergodic, with the field $\psi_l(t)$ forming strongly localized self-trapped large-amplitude excitations which are persisting on a background of delocalized waves [17, 27–29]. The self-trapped field part appears to condense in a way such that the remaining background field part can evolve at an infinite temperature in a Gibbs regime. Since the wave packet spreading process is a non-equilibrium one, we are lacking a clear condition to determine the precise amount of condensed norm. We can only speculate (and observe) that a background fraction of the wave packet will continue to be de-

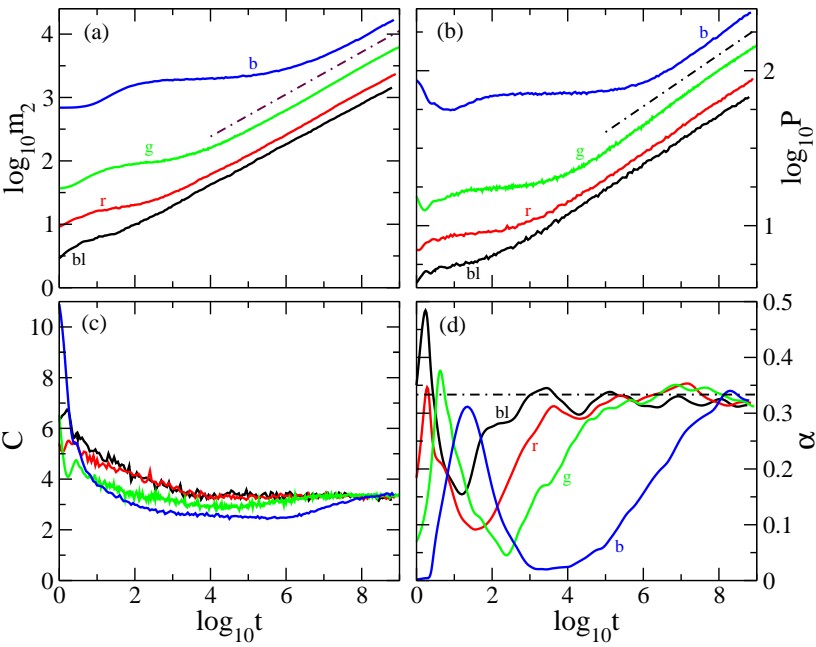

Figure 3: Density resolved weak chaos wave packet spreading. $h = 0$, $a < d$. (a) $\log_{10} m_2(t)$ vs. $\log_{10} t$. Dashed dotted line: $m_2(t) \sim t^{1/3}$. (b) $\log_{10} P$ vs. $\log_{10} t$. Dashed dotted line: $P(t) \sim t^{1/6}$. (c) $C$ vs. $\log_{10} t$. (d) $\alpha$ vs. $\log_{10} t$. Horizontal line: $\alpha = 1/3$. The parameters are: $a = 0.0025$ and $W = 2$ for (b)lue line, $a = 0.05$ and $W = 4$ for (g)reen line, $a = 0.245$ and $W = 6$ for (r)ed line, $a = 0.9$ and $W = 8$ for (bl)ack line. The system size $N = 2^{10}$, the number of disorder realizations is 200.

pleted but yet spread possibly with characteristics of strong and weak chaos. The evolution of the norm density of a wave packet in the self-trapping regime is plotted versus space and time in Fig.2(d). We clearly observe a remaining strongly localized self-trapped part of the wave packet at the origin, which is much more localized than its Anderson localization counterpart in Fig.2(a). The background part of the wave packet instead spreads qualitatively similar to the weak (and strong) chaos case Fig.2(b). Consequently, the second moment of the wave packet grows in time due to the spreading of the background part (red line Fig.5(a)). At the same time, the participation number drops as time increases, which indicates a slow but steady formation of self-trapped excitations on very few lattice sites (red line Fig.5(b)). The compactness index evolution (red line Fig.5(c)) confirms these findings very well. Finally, the derivative $\alpha(t)$ (red line Fig.5(d)) indicates that the background wave packet part may reach asymptotic weak chaos characteristics at times which are not accessible by our computational resources.

The wave packet spreading in both the Lifshits phase and the self-trapping regime is characterized by a substantial slowing down from the subdiffusive spreading as observed for weak and strong chaos. At the same time, the self-trapping enforces highly localized almost single site excitations to be formed, at variance to the Lifshits phase dynamics where the wave packet structure resembles the Anderson localization case.

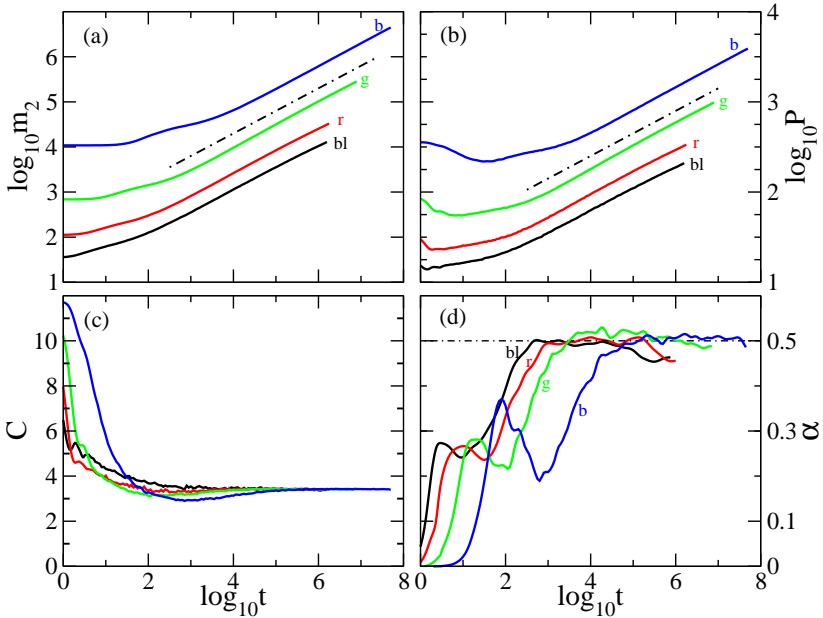

Figure 4: Density resolved strong chaos wave packet spreading. $h = 0$, $a > d$. (a) $\log_{10} m_2(t)$ vs. $\log_{10} t$. Dashed dotted line: $m_2(t) \sim t^{1/2}$. (b) $\log_{10} P$ vs. $\log_{10} t$. Dashed dotted line: $P(t) \sim t^{1/4}$. (c) $C$ vs. $\log_{10} t$. (d) $\alpha$ vs. $\log_{10} t$. Horizontal line: $\alpha = 1/2$. The parameters are: $a = 0.047$ and $W = 1$ for (b)lue line, $a = 0.19$ and $W = 2$ for (g)reen line, $a = 0.4$ and $W = 3$ for (r)ed line, $a = 0.79$ and $W = 4$ for (bl)ack line. The system size $N = 2^{13}$, the number of disorder realizations 200 for $W = 1$, 500 for $W = 2$, 750 for $W = 3$, 1000 for $W = 4$.

## 8   Conclusion

We have analyzed energy and norm density resolved wave packet spreading studies in the disordered Gross-Pitaevskii (GP) lattice. We managed to confine energy density fluctuations, which were not controlled in previous studies. We mapped the locations of the GP regimes of weak and strong chaos sub-diffusive spreading in the two-dimensional density control parameter space and observed strong chaos spreading over several decades. A number of qualitatively different wave packet spreading outcomes were identified. Positive energy densities are doomed to bring the wave packet into a non-Gibbs regime with potential fragmentation of the packets into a self-trapped condensate part and an infinite temperature background capable of spreading infinitely. One of the intriguing quantities is the ratio of the norm in the two field components and its asymptotic time dependence. Will the self-trapped component take over the entire wave packet norm at large enough times, or will some finite remain in the infinite temperature background? Zero energy densities keep the wave packet in the Gibbs regime and may lead to the entire packet heating up to infinite temperatures upon infinite spreading. Negative energy densities in the Gibbs regime can bring the wave packet closer to the ground state, and therefore zero temperatures upon spreading. Finally, initializing the wave packet in the Lifshits regime shows strong suppression of subdiffusive spreading. But even in this case, we notice a speedup of the spreading process with some potential fragmentation of the wave packet. It appears that there are a number of interesting and hard open problems to be

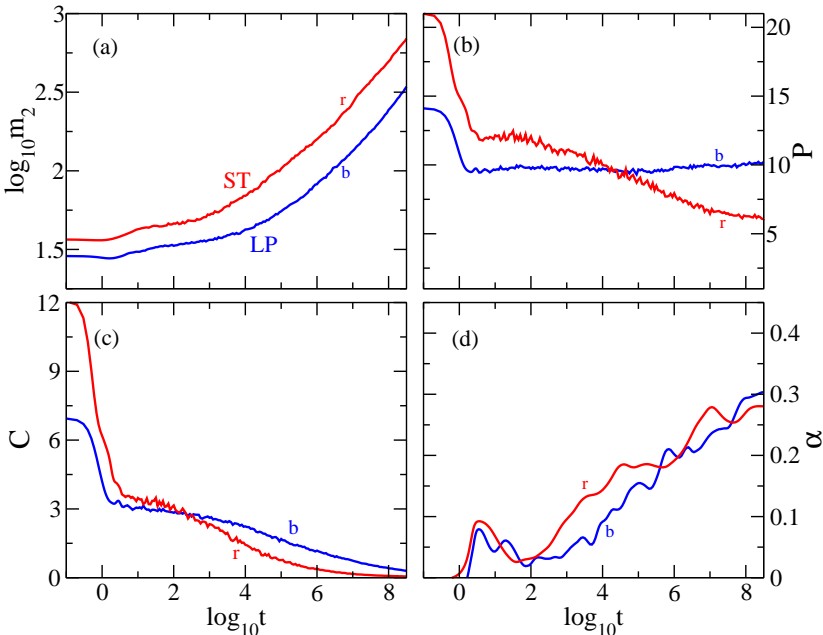

Figure 5: Density resolved wave packet spreading in the Lifshits phase (b)lue lines, and the self-trapping regime (r)ed lines. (a) $\log_{10} m_2(t)$ vs. $\log_{10} t$. (b) $P$ vs. $\log_{10} t$. (c) $C$ vs. $\log_{10} t$. (d) $\alpha$ vs. $\log_{10} t$. Lifshits phase: $h = -0.277$. Self-trapping regime: $h = 0.277$. Other parameters: $W = 4$, $a = 0.1$, $N = 2^{10}$, averaging over 200 disorder realizations for LP, 100 for ST.

addressed in future work.

## Acknowledgements

We are grateful to S. Gündoğdu, I. Vakulchyk, A. Cherny, M. Fistul, A. Andreanov, L. Toikka and J.D. Bodyfelt for insightful discussions. We acknowledge the contribution of NZ eScience Infrastructure (NeSI) high-performance computing facilities. X.Y. gratefully acknowledges the hospitality of the IBS Center of Theoretical Physics of Complex Systems, while part of this work was accomplished.

**Funding information** This work was supported by the Institute for Basic Science (Project number IBS-R024-D1). X.Y. acknowledges the support from NSAF through grant number U1930403.

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
