# Peer review of "Density Resolved Wave Packet Spreading in Disordered Gross-Pitaevskii Lattices"

_SciPost Physics_

## Round 2 · Referee Report · Anonymous (Referee 1) · 2020-7-12

Report

In this paper, the authors are revisiting the problem of wave packet spreading in a disordered nonlinear Schroedinger chain. This problem has attracted a lot of interest in the last 10-12 years, including several publications by the same research group. The novel aspect of the present study is the control on both conserved quantities of the system, norm and energy. In earlier works, the initial conditions had only a given norm, while the energy was different in different disorder realisations (averaged over), while here the system's behaviour is studied for given norm an energy.

The most important result of the paper is the observation of the strong chaos regime, characterised by the exponent alpha = 1/2 in contrast to the weak chaos with alpha = 1/3. This regime was not found in earlier studies of the nonlinear Shroedinger chain precisely because the energy of each initial condition was not well controlled. It was seen only in the Klein-Gordon chain which has only one conserved quantity (the energy). This is an interesting result which definitely deserves a publication.

Unfortunately, the authors only present evidence for the existence of the strong chaos regime, but not a systematic study of the conditions where this regime occurs. It was argued in earlier publications that strong chaos should occur when the norm density exceeds the mean frequency spacing in the localisation volume. This could be checked numerically (a) by looking at different initial conditions, and (b) by looking at the time when the strong chaos crosses over to the weak chaos which should inevitably occur during the wave packet expansion, but no systematic quantitative study is presented. As far as the energy is concerned, no condition is given at all, even as a conjecture. For this reason, the paper seems to me more suitable for publication in Scipost Physics Core rather than in Scipost Physics.

I have a couple of minor comments, mostly concerning the presentation of technical details.

1. Could the authors explain in more detail their reasons for choosing the method of minimising the Hamiltonian in Sec. 3? I always thought that the most standard way would be to use some gradient-based method. Also, I would naively expect the authors' method to be efficient in minimising local configurations, but not in equilibrating populations of two distant Lifshitz minima. I am not questioning the authors' results, which are correctly verified by the local chemical potential test, but I guess many readers (including myself) would appreciate some idea of why this method was chosen.

2. In Sec. 7, I am a bit surprised by the authors' statement that in the non-Gibbs regime the norm density in the spreading fraction cannot be determined and the regime cannot be characterised. Why is it impossible to determine the norm in the self-trapped fraction and subtract it? Also, the wave packet second moment growth is supposed to characterise the spreading fraction, since the self-trapped fraction is frozen, isn't it? The exponent in Fig. 5(d) does not seem to saturate. Is it possible to go to longer times and see if it goes beyond 1/3 (maybe by choosing different initial conditions)?

---

## Round 2 · Referee Report · Anonymous (Referee 2) · 2020-7-18

Report

The paper is devoted to a study of spreading of an initially localized wavepacket in a DANSE (disordered Anderson nonlinear Schroedinger lattice model). It follows a framework of previous publications by calculating the growth rate of the second moment of the field and of the participation number. An addition to previous studies is not an extension of the considered time interval (it is about 10^9), but in a separate consideration of regimes for different values of the parameters of the initial wavepacket - its energy and its norm. Correspondingly, behaviors at least at initial stage discriminated. This contribution to previous studies definitely deserves publication. The paper is clearly written, and I have only one suggestion: place the "trajectories"on the plane (a,h) for the scenaria presented in figs 3-5 also on Fig 2. Or, if this is difficult, place at least the initial conditions for the runs at W=4 as markers on Fig. 2, this will help readers to arrange the simulations to the theoretical considerations.

---

## Editorial Decision

resubmitted